# Tooth-Specific *Streptococcus mutans* Distribution and Associated Microbiome

**DOI:** 10.3390/microorganisms10061129

**Published:** 2022-05-31

**Authors:** Márcia Dinis, William Traynor, Melissa Agnello, Myung-Shin Sim, Xuesong He, Wenyuan Shi, Renate Lux, Nini Chaichanasakul Tran

**Affiliations:** 1Section of Pediatric Dentistry, School of Dentistry, University of California, Los Angeles, CA 90095, USA; marciadinis@ucla.edu (M.D.); wtraynor@gmail.com (W.T.); 2Section of Oral Biology, School of Dentistry, University of California, Los Angeles, CA 90095, USA; agnellom@gmail.com (M.A.); xhe@forsyth.org (X.H.); wshi@forsyth.org (W.S.); 3Division of General Internal Medicine and Health Services Research, Department of Medicine Statistics Core, University of California, Los Angeles, CA 90095, USA; msim@mednet.ucla.edu; 4The Forsyth Institute, Microbiology, Cambridge, MA 02142, USA; 5Section of Periodontics, School of Dentistry, University of California, Los Angeles, CA 90095, USA; rlux@dentistry.ucla.edu

**Keywords:** *Streptococcus mutans*, dental caries, posterior teeth, dentition stages, tooth-associated microbiome

## Abstract

Dental caries is multifactorial and polymicrobial in nature and remains one of the most common oral diseases. While caries research has focused on *Streptococcus mutans* as the main etiological pathogen, its impact at the tooth level is not fully understood. In this cross-sectional study, the levels and distribution of *S. mutans* in the posterior teeth at different dentition stages were investigated along with the corresponding tooth-specific microbiome. Occlusal plaque samples of 87 individual posterior teeth were collected from thirty children in three dentition stages (primary, mixed, and permanent). The *S. mutans* levels in the occlusal plaque of individual posterior teeth were quantified with qPCR, and those with preferential colonization were selected for tooth-specific microbiome analysis using 16S rRNA sequencing. Results: Quantification of *S. mutans* levels in the occlusal plaque confirmed the preferential colonization on the first primary and permanent molars. These teeth were selected for further tooth-specific microbiome sequencing, as they also displayed high caries experience. There were significant differences in the relative abundance of the four most abundant genera: *Neisseria*, *Streptococcus*, *Rothia,* and *Veillonella*. Furthermore, the tooth-level caries experience was correlated with a reduction in the microbiome diversity. Analyzing the different tooth-associated microbial communities, distinct tooth-specific core microbiomes were identified. Conclusions: Our findings suggest that caries susceptibility at the tooth level, depending on tooth type and dentition stage, is influenced by individual species as well as plaque community.

## 1. Introduction

Dental caries is the most prevalent chronic oral infectious disease, which affects over 40% of children and adolescents in the US and is a global oral health burden [1,2]. Caries-associated pain and infection in childhood often have life-long consequences for overall health and well-being [3]. The etiology of this multifactorial disease involves complex interactions between the various genetic, microbial, behavioral, and environmental factors [4]. In children, the oral cavity is a dynamic environment with several transitions between the various dentition stages (primary, mixed, and permanent) [5]. Oral health or disease between these stages are strongly correlated, as caries experience in primary teeth often influences the caries outcome in permanent dentition [6]. The caries experience in primary posterior teeth has been established as a reliable predictor for the subsequent decay in permanent dentition [7,8]. These posterior teeth have an increased caries susceptibility due to their distinctive pit and fissure occlusal topography, arch position as well as eruption status that provides a favorable niche for bacterial colonization [9,10,11].

*Streptococcus mutans*, a major etiological agent for caries, is prevalent on these occlusal surfaces and has been explored as a risk predictor of the disease [12,13,14]. A number of studies reported a direct association between the levels of *S. mutans* and the higher probability of caries experience [15,16,17,18,19,20], while others noted a lack of correlation [21,22]. In recent years, the paradigm of *S. mutans* as the principal cariogenic pathogen has shifted with increasing evidence supporting the polymicrobial nature of dental caries. Especially in children, the oral cavity is a highly heterogeneous ecological system. The transition between dentition stages impacts the microbial habitats, leading to a significant shift in the oral microbiota diversity and composition [23,24,25,26,27].

To date, the correlation between the oral microbiome and caries in various dentition stages has been examined using saliva [28,29,30] or pooled dental plaque samples from multiple tooth surfaces [22,29,31,32,33,34,35]. However, only a few studies have focused on the microbial profiles of the tooth, using pooled multiple occlusal surfaces [31,36,37] or site-specific supragingival plaque [35,38], as well as interdental biofilm [39], and found substantial differences in the supragingival dental plaque microbiome among tooth surfaces and caries experience of children. In addition to *S. mutans*, other species such as *Catonella* [40,41,42] and *Veillonella* were reported in active caries, which supports the ecological nature of caries etiology.

While additional species may play a role in dental caries development, a considerable amount of research has established *S. mutans* as a primary cariogenic pathogen. Detailed analysis of the preferential localization of *S. mutans* at the tooth level in different dentition stages and its impact on the tooth-specific microbiome are not fully explored. In this cross-sectional study, we evaluated the levels and distribution of *S. mutans* in the occlusal plaque of posterior teeth throughout the dentition stages and the associated tooth-specific microbiome in caries-active children.

## 2. Materials and Methods

### 2.1. Ethics

The study was reviewed and approved by the University of California Los Angeles Institutional Review Board (#16-000289). Parents or legal guardians of all participants signed written informed consent prior to the initiation of the study. The study was conducted in accordance with the declaration of Helsinki for Biomedical Research Involving Human Subjects.

### 2.2. Recruitment and Sampling

Study participants were recruited from the pediatric patient population of the University of California, Los Angeles (UCLA) Children’s Dental Center during their periodic oral examination. Thirty healthy children were enrolled and stratified into three groups of ten subjects per group according to their age and dentition stage: primary (2–5 years), mixed (6–10 years), and permanent dentition (11–15 years). The participants’ wide age range was necessary to capture posterior tooth types from different dentition stages. Each participant completed a brief questionnaire, which provided demographic information, oral hygiene habits, and treatment history. Participants were eligible based on the following inclusion criteria: healthy children with caries experience (dmft/DMFT score > 0) who were not taking any medication and had no antibiotic use within the last six months. Participants were excluded from the study if they had generalized rampant caries, periodontitis, halitosis, open sores or ulcerations, chronic systemic diseases, reduced saliva production, or other medical conditions that are known to influence the oral cavity’s microbiome.

Oral clinical evaluation was performed by a trained clinical investigator. The caries experience of each posterior tooth within the different dentition stages: primary first (D’s) and second (E’s) molars, permanent first (4’s) and second (5’s) pre-molars, permanent first (6’s), and second (7’s) molars, was evaluated using decayed, missing, and filled tooth (DMFT) index, according to the criteria proposed by the World Health Organization [43].

The dental plaque collection was standardized, and all participants were asked to refrain from any oral hygiene activity, eating, and drinking in the morning before oral sampling. A total of 87 samples were collected, details per tooth type are described in Table 1 and Table A1, from the individual occlusal surfaces of specific posterior teeth (D’s, E’s, 4’s, 5’s, 6’s, and 7’s) using a sterile Pikster^®^ #3 brush per tooth type (Piksters, Macksville, Australia). Samples derived from the same tooth type in all four quadrants were pooled (four-quadrant sample) into sterile 1.5 mL micro-centrifuge tubes (Eppendorf, Hamburg, Germany) containing PBS and glycerol. After collection, all samples were immediately frozen and stored at −80 °C.

### 2.3. DNA Extraction

Total genomic DNA was extracted from the 87 occlusal plaque samples using the Epicenter MasterPureTM DNA purification kit (Lucigen, Middleton, WI, USA), following the manufacturer’s instructions with modifications. Briefly, plaque samples were subjected to mechanical grinding with glass beads followed by lysozyme treatment for two hours at 37 °C [44]. The DNA quantity and quality were measured with NanoDrop 2000 (ThermoFisher Scientific, Waltham, MA, USA). The extracted DNA was stored at −20 °C until further use.

### 2.4. Quantitative Polymerase Chain Reaction (q-PCR)

The total bacteria and *S. mutans* levels in the 87 occlusal plaque samples were assessed using Real-Time Quantitative Polymerase Chain Reaction (qPCR). Primers targeting the conserved region of the 16S rRNA gene and *S. mutans*-specific primers targeting glucosyltransferase B encoding gene were used for the total bacteria [45] and *S. mutans* [46] detection, respectively. 

The qPCR reactions were performed in 96-well plates (Biorad, Hercules, CA, USA) using Bio-Rad iCycler Thermal Cycler with iQ5 Multicolor Real-Time PCR Detection System (BioRad iQ 5). Briefly, the reaction mix comprised of 0.5 µM of each primer with 1X SYBR Green Master Mix (BioRad) and 15 ng of DNA in a reaction volume of 20 µL. The qPCR program consisted of an initial denaturation for 5 min at 95 °C, followed by 40 cycles of denaturation at 95 °C for 30 s, annealing at 64 °C for 30 s, and extension s at 72 °C for 30 s. The qPCRs were performed with three technical replicates for each sample. The iQ5 Optical System Software generated quantification cycle values (Cq) were analyzed along with melting point data. A standard curve was generated from a serial dilution of *S. mutans* reference strain (UA140) DNA.

### 2.5. Microbial Community Analysis

A total of 36 occlusal plaque samples: primary Dprim’s (n = 10), Dmixed’s (n = 8), and permanent 6mixed’s (n = 8), 6perm’s (n = 10) first molars were selected for sequencing (Table 1). Briefly, genomic DNA was used to amplify the V4 region of the 16S gene and sequenced on the Illumina Miseq sequencer system at the UCLA Microbiome Core, using a 150 bp paired-end protocol. The sequencing data were analyzed using QIIME (Quantitative Insights into Microbial Ecology) version 1.9.1 [47]. Sequences were clustered into operational taxonomic units (OTUs) using UCLUST [48], and the taxonomy was assigned with the Human Oral Microbiome Database (HOMD) [49] as a reference. A total of 3,761,710 high-quality sequences from thirty-six samples, with a mean of 104,491 ± 2732, were generated after processing the data obtained from 16S rRNA gene sequencing. For alpha diversity, OTU tables were rarefied to 23,000 reads, Shannon index was calculated [50]. For beta diversity, UniFrac distances were calculated, followed by principal coordinates analysis (PCoA). The OTUs lists generated for the several core analyses were compared using Venn diagram software packages [51].

### 2.6. Power Analysis

Comparing between D’s and 6’s, with the sample size of 10 per group, we had 80% power to detect significant differences at a 5% significance level using analysis for the repeated measures assuming a correlation of 0.7 between the teeth from the same subject. This calculation assumed that the mean difference of 2.8 would be detected (the means in caries experience: 3.3 for D’s vs. 0.5 for 6’s) with the standard deviation of 1.7.

### 2.7. Statistical Analysis

The statistical tests were performed using GraphPad Prism (version 8.0.0, GraphPad Software, San Diego, CA, USA), and the statistical significance was defined as follows: * *p* < 0.05, ** *p* < 0.005, and *** *p* < 0.0005. The demographic and *S. mutans* distribution data analysis was assessed by one-way ANOVA or Kruskal–Wallis test for comparing more than 2 independent groups. Wilcoxon signed-rank test was performed in comparing dmft/DMFT scores and *S. mutans* levels that were collected within the same subjects. The Spearman correlation analysis was performed between the demographic and oral hygiene parameters. The association between dmft/DMFT scores and *S. mutans* levels was evaluated by linear regression. The tooth microbiome statistical analysis was determined with the Kruskal–Wallis test, applying the false discovery rate to correct for multiple comparisons. Differences in weighted Unifrac distances between the groups were analyzed with analysis of similarity (ANOSIM).

## 3. Results

### 3.1. Study Demographics

A total of thirty pediatric participants were enrolled in this cross-sectional study and were divided into three groups according to their age and type of dentition. In brief, the gender distribution was 17 females (57%) and 13 males (43%). The study population comprised 87% Hispanic and 13% non-Hispanic with an average age of 3.9 (±1.0), 8.6 (±2.1), and 12.4 (±1.8) for primary, mixed, and permanent dentition groups, respectively. The frequency of their oral hygiene habits, such as professional cleaning, their tooth brushing and flossing routine, and mouth rinse usage, were similar between the groups. All subjects reported not using fluoride supplements.

### 3.2. Caries Experience and Streptococcus mutans Localization in Different Types of Posterior Teeth

The caries status of individual teeth per type of posterior tooth (D’s, E’s, 4’s, 5’s, 6’s, and 7’s), a total of 87 samples, was assessed by the dmft/DMFT index and revealed that the caries experience of the first primary molars (D’s) was significantly higher in comparison to all other teeth, except for the primary second molars (E’s). Among the permanent teeth, the first permanent molars (6’s) exhibited the highest caries experience (Figure 1a). Although not statistically significant, a preferential localization of *S. mutans* was observed in the primary and permanent first molars, the posterior teeth that erupt first within each dentition (Figure 1b). Even though linear regression analysis demonstrated that *S. mutans* levels and tooth-specific caries experience were not directly correlated (Figure 1c), it is important to highlight that D’s and 6’s appeared to have higher caries experience and harbor more *S. mutans*.

### 3.3. Community Profiling of D’s and 6’s

Next, we investigated the influence of *S. mutans* colonization on the local tooth microbiome of D’s and 6’s in the different dentition stages (Dprim, Dmixed, 6mixed, 6perm), a total of 36 posterior teeth. The alpha diversity Shannon index revealed that the *S. mutans* levels (Figure 2a) did not affect the tooth microbial diversity. In contrast, a significant decrease in diversity was observed on teeth with high caries compared to caries-free teeth (Figure 2b) and on Dmixed in comparison to 6mixed, when posterior teeth per dentition were compared (Figure 2c).

Further, microbial community changes were examined using weighted Unifrac distances according to S. mutans levels, tooth-specific caries experience, and tooth type per dentition (D’s, 6’s) (Figure 2d–f). The *S. mutans* levels were categorized into two groups: low (<1%) and high (≥1%) (unpublished data) based on qPCR-quantification, significantly clustered according to S. mutans percentage (Figure 2d). For the tooth-specific caries experience, an apparent trend was observed for the teeth with high DMFT but was not statistically significant (Figure 2e). Additional beta diversity analyses considering the tooth type in the different dentition stages did not reveal a specific pattern (Figure 2f).

The local microbial composition of D’s and 6’s in different dentition stages at the genus level revealed a total of forty-five genera, but for further analysis, only those with a mean abundance value of ≥ 0.1% were included (Figure 3a). Of the 25 of these genera, significant differences between the tooth types were only observed for *Streptococcus, Neisseria, Rothia*, and *Veillonella* (Figure 3b). While the abundance of Neisseria was significantly higher in the Dprim’s (20.3% ± 7.3) in comparison to the 6perm’s (13.6% ± 11.5), the relative abundance of Rothia and Streptococcus were highest in 6perm’s (13.5% ± 8.7) and Dprim’s (26.5% ± 7.2), respectively, and differed significantly from the other tooth types. Additionally, significantly higher levels of Veillonella were observed in 6mixed’s (9.4% ± 1.4) in comparison to Dmixed’s (2.4% ± 1.3) (Figure 3b).

### 3.4. Core Microbiome of D’s and 6’s

The different tooth-associated microbial communities were analyzed for their respective shared and unique genera. In all posterior teeth (D’s and 6’s), the core microbiome consisted of twenty-eight genera (80.0%), including the highly prevalent genera, *Streptococcus*, *Neisseria*, *Rothia*, and *Veillonella*, among others (Figure 4a).

When comparing between D’s and 6’s besides the core genera, seven genera (*Catonella*, *Eikenella, Klebsiella*, *Lachnoanaerobaculum*, *Peptococcus*, *Peptostreptococcus*, and *Tannerella*) were specific to the 6’s tooth type. Further analysis of the D’s and 6’s based on their dentition stages revealed that besides a similar core microbiome of 23 genera (66%), specific genera were associated to particular tooth depending on dentition (Figure 4b). The Dprim’s and Dmixed’s had only one genus (*Cardiobacterium*) in common, while 6mixed’s and 6perm’s shared three genera, *Klebsiella*, *Lachnoanaerobaculum*, and *Tannerella.* Interestingly, four genera were found to be distinct to 6’s according to the dentition stage; Catonella and Eikenella were specific to 6mixed’s, while *Peptococcus* and *Peptostreptococcus* were unique to 6perm’s.

## 4. Discussion

*Streptococcus mutans*, the principal etiological agent of dental caries, is frequently employed for caries risk assessment [12,52], even though the levels are not always correlated with caries experience [21,22,53,54]. Despite evidence supporting a higher prevalence of *S. mutans* in posterior teeth [55], the pattern of *S. mutans* levels in individual posterior teeth across dentition stages is not fully characterized. Therefore, in this cross-sectional study, we investigated the levels of *S. mutans* and caries experience in individual types of posterior teeth (D’s, E’s, 4’s, 5’s, 6’s, and 7’s), the localization pattern of this bacterium, and explored how *S. mutans* influences the local tooth microbiome.

This study indicates a differential caries susceptibility and *S. mutans* localization at the tooth level. The highest caries scores were observed in the first primary molars (D’s) for the primary dentition and the first permanent molars (6’s) for the permanent dentition (Figure 1a). These results are in agreement with other studies, albeit using a different caries index system [56] and in older populations [57]. In particular, the posterior teeth that erupted first and resided longer, Dprim’s and Dmixed’s, displayed the higher caries scores. Interestingly, although not statistically significant, *S. mutans* appears to favor colonization of D’s and 6’s, the first molars within each dentition stage (Figure 1b), with 6’s having the highest *S. mutans* levels among the evaluated teeth. The apparent preferential localization of *S. mutans* could be explained by the fact that these teeth are the first to erupt and are located more anteriorly with a distinct pit and fissure morphology that may promote efficient *S. mutans* adhesion and plaque accumulation [36,52,55].

To explore the influence of *S. mutans* and caries experience on the local tooth microbiome, we selected representative teeth (Dprim’s, Dmixed’s, 6mixed’s, and 6perm’s) for sequencing analysis. Overall, species diversity was similar between all the teeth, independent of *S. mutans* levels. On the contrary, a previous study suggested that the dominance of *S. mutans* may decrease community diversity [38]. Regarding caries experience, significantly reduced microbial diversity was observed on teeth with high caries compared to caries-free teeth. This observation was similar to other studies that described decreasing microbial diversity with the severity of caries [31,37,38]. Interestingly, in this study, significantly higher species abundance was observed in the 6mixed’s than in the corresponding Dmixed’s, highlighting a potential tooth-level impact on species diversity.

At the genus level, the four most abundant genera were identified, including *Streptococcus, Neisseria*, *Rothia*, and *Veillonella*, which were also reported in previous investigations [24,32]. The members of the *Streptococcus* and *Neisseria* genus, predominant in this study, are early colonizers and often dominate the dental plaque in the primary dentition. In concurrence with our findings, previous studies reported that *Streptococcus* and *Neisseria* levels tend to decrease as dentition matures, and late colonizers further contribute to diversity [58,59]. In contrast, the genus *Rothia* increased from primary to permanent dentition, with the highest levels in 6perm’s, indicating that this genus is present in all dentitions with higher abundance in the permanent dentition [24,25].

Moreover, this genus has been associated with either health [28,30,42] or caries [22,29,33], depending on the species. Although not significantly different, the 6’s had the highest caries experience in comparison to the other permanent teeth (4’s, 5’s, and 7’s). Previously, a higher abundance of *Veillonella* has been associated with permanent dentition [59]. However, in this study, low levels of *Veillonella* were observed in 6 mixed’s, suggesting an ongoing transition in the oral microbiome.

While transitioning through dentition stages is a dynamic process, part of the microbiome remains constant [23,25]. To better understand the microbe–habitat relationship, the ‘core microbiome’ of D’s and 6’s in caries-active children was evaluated. Within the core microbiome of 28 genera observed between D’s and 6’s (Figure 4b), 19 genera were consistent with those reported previously for the core microbial community in healthy children with a similar age range [23,24]. Additional genera were detected in our study, which focused on caries-active children transitioning through various dentition stages.

Further investigation into the dentition-specific cores of D’s and 6’s revealed that Dprim’s and Dmixed’s shared the genus *Cardiobacterium*, and 6mixed’s and 6perm’s shared three genera, *Klebsiella*, *Lachnoanaerobaculum*, and *Tannerella.* While *Cardiobacterium* has been associated with healthy primary dentition [40], *Klebsiella* is reported in permanent dentition of healthy individuals [35]. On the other hand, *Tannerella* was only present in the saliva of caries active children with mixed dentition, suggesting the effect of both caries and transition of dentition stages [35]. Furthermore, the presence of *Peptococcus* and *Peptostreptococcus* only in 6perm’s suggests the association of these genera with the mature oral microbiome. Similar to the observations of this study, previous studies with healthy individuals identified *Peptococcus* and *Peptostreptococcus* in the later stage of primary and permanent dentitions [26]. In this study, the sequencing of the V4 region of the 16S gene did not allow the microbiome analysis at the species level. Although 16S rRNA gene sequencing can provide adequate phylogenetic information to identify the bacteria at the species level, for some taxa, such as different streptococcal species, the identification is limited. In the future, it would be interesting to explore within the identified genera which species are determinant to the tooth-specific microbiome in the caries context.

This cross-sectional study highlights the clinical importance of individual tooth susceptibility to dental caries. The strength of our study was the tooth-type specific plaque investigation compared to other studies, which utilized pooled occlusal [31,36,37] and supragingival [35,38] dental plaque or saliva [28,29,30]. This study revealed that *S. mutans* preferentially colonized D’s and 6’s and detected differences at the phylogenetic level in the tooth-specific microbiome associated with dental caries throughout dentition stages. Our study design attempted to unravel the microbiome shifts through dentition stages which would be challenging to accomplish in a longitudinal study. Building on our pilot study, which indicated microbial shifts occurring between the different dentition stages, future studies with an increased number of samples per tooth type and the inclusion of all the posterior teeth for microbiome analysis would contribute valuable data to understanding the tooth-level and dentition-stage-dependent caries susceptibility.

## Figures and Tables

**Figure 1 microorganisms-10-01129-f001:**
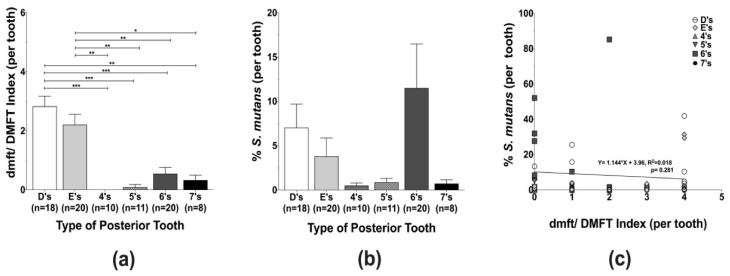
Caries experience *and Streptococcus mutans* levels quantification among posterior tooth in different dentition stages. (**a**) The individual posterior tooth (D’s, E’s, 4’s, 5’s, 6’s, and 7’s) caries status was assessed using decayed, missing, and filled tooth (dmft/DMFT) criteria, and the figure represents the average caries experience per tooth type. (**b**) *S. mutans* levels were evaluated by qPCR using *S. mutans*-specific primers (Table 1). (**c**) Linear regression analysis of *S. mutans* levels and dmft/DMFT index. Differences in significance between groups were analyzed using one-way ANOVA followed by Kruskal–Wallis multiple comparisons tests; * *p* < 0.05, ** *p* < 0.005, and *** *p* < 0.0005.

**Figure 2 microorganisms-10-01129-f002:**
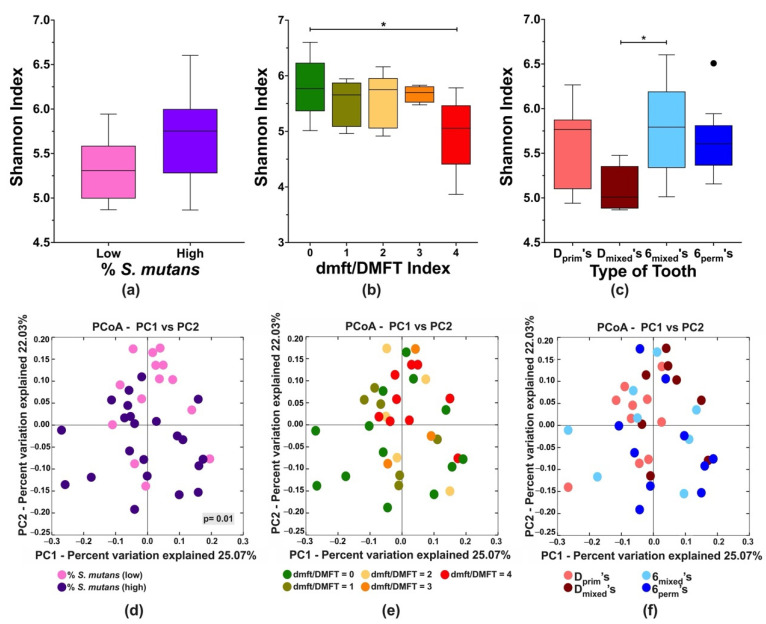
Alpha and beta diversity analyses of the microbial community associated with first primary and permanent molars within the different dentition stages (Dprim’s, Dmixed’s, 6mixed’s, and 6perm’s). Alpha diversity was analyzed using Shannon index (upper panel, (**a**–**c**)), and beta diversity was calculated by principal coordinates analysis (PCoA) of weighted Unifrac distances (lower panel, (**d**–**f**)). The analyses were performed based on the following parameters: (**a**,**d**) *S. mutans* levels, low (% < 1%) and high (% ≥ 1%); (**b**,**e**) posterior tooth dmft/DMFT index; and (**c**,**f**) tooth type per dentition. The differences in the significance of the groups were performed by one-way ANOVA followed by Kruskal–Wallis multiple comparisons tests; differences in unweighted Unifrac distances between the groups were analyzed with analysis of similarity (ANOSIM); * *p* < 0.05. The black dot in Figure 2c represents an outlier.

**Figure 3 microorganisms-10-01129-f003:**
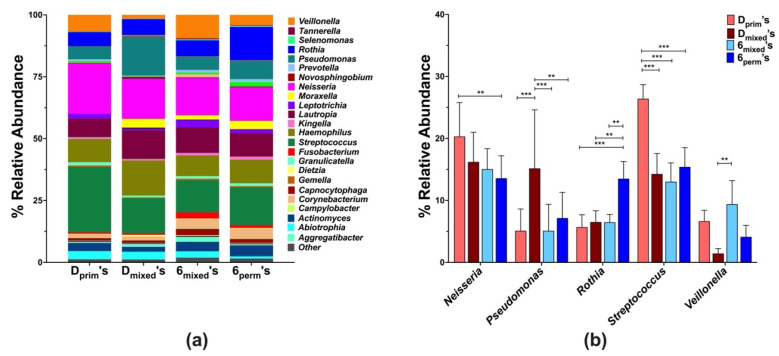
Genus level analysis of plaque communities associated with first primary and permanent molars within the different dentition stages (Dprim’s, Dmixed’s, 6mixed’s, and 6perm’s). Relative abundances at the genus level are shown for microbial profiles (**a**) and selected genus (**b**). The statistical analysis of the tooth microbiome at the genus level was determined with the Kruskal–Wallis test, controlling the false discovery rate to correct for multiple comparisons. Statistical significance between groups was evaluated by two-way ANOVA followed by Tukey’s multiple comparisons tests; ** *p*< 0.005, and *** *p* < 0.0005.

**Figure 4 microorganisms-10-01129-f004:**
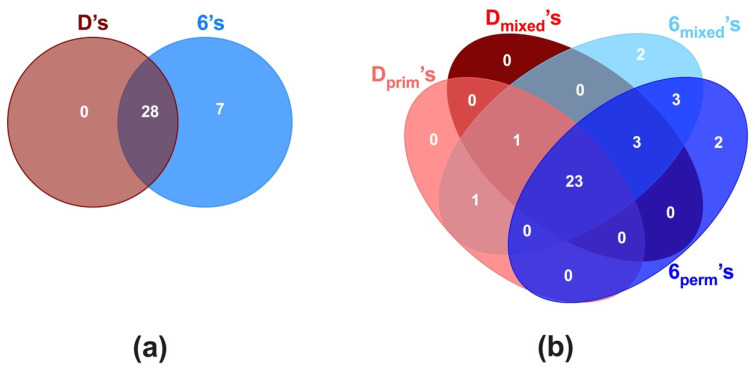
Core microbiome at the genus level of first primary and permanent molars (Dprim’s, Dmixed’s, 6mixed’s, and 6perm’s). The analyses of the first primary and permanent molars teeth core microbiome are displayed as Venn diagrams, based on (**a**) type of tooth and (**b**) tooth type per dentition.

**Table 1 microorganisms-10-01129-t001:** Summary of the samples collected in the study. The occlusal plaque of individual posterior teeth (87 samples) was collected from the thirty enrolled participants, and the samples were stratified into three groups according to the dentition stage. The caries experience of each tooth was evaluated using dmft/DMFT index; the respective average per tooth type is indicated in the table.

		Dentition Stage		
		Primary (2–5 y) (n = 10 Participants)	Mixed (6–10 y) (n = 10 Participants)	Permanent (11–15 y) (n = 10 Participants)
		dmft/DMFT Index		
Type of Posterior Tooth	Code	dmft	dmft/DMFT	DMFT
Primary 1st molar	D’s	2.1 ± 1.7 (n = 10)	3.3 ± 1.1 (n = 8)	
Primary 2nd molar	E’s	2.0 ± 1.8 (n = 10)	2.1 ± 1.5 (n = 10)	
Permanent 1st pre-molar	4’s		0.0 ± 0.0 (n = 1)	0.0 ± 0.0 (n = 9)
Permanent 2nd pre-molar	5’s		0.0 ± 0.0 (n = 1)	0.1 ± 0.3 (n = 10)
Permanent 1st molar	6’s		0.3 ± 0.7 (n = 10)	0.8 ± 1.1 (n = 10)
Permanent 2nd molar	7’s		0.0 ± 0.0 (n = 1)	0.3 ± 0.5 (n = 7)

Code—tooth nomenclature using Palmer notion; dmft/DMFT—decay missing filled tooth index, per individual tooth type. For the sequencing experiments, we selected (highlighted in grey): primary first molars from primary (D_prim_’s, n = 10) and mixed (D_mixed_’s, n = 8) dentitions; permanent first molars from mixed (6_mixed_’s, n = 8) and permanent (6_perm_’s, n = 10) dentitions.

## Data Availability

The data that support the findings of the study are available from the corresponding authors upon request.

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
