# Peer review of "Tooth-Specific Streptococcus mutans Distribution and Associated Microbiome"

_microorganisms, 2022, doi:10.3390/microorganisms10061129_

Round 1

Reviewer 1 Report

I am honestly very happy to see a paper like this in the Microorganisms journal. It is the best one that I have reviewed for it thus far. I see no flaws in the design or methodology of the study and the paper is concisely written and the conclusions are warranted. I don't usually do this but I am fine with this being published in its current form. 

Author Response

Dear Reviewer 1,

On behalf of all authors, we thank you for your time and effort in reviewing our manuscript. We really appreciate your comments and are elated that you warranted our work to be suitable for publication.

Thank you once again for your time and effort in reviewing our manuscript.

Nini Tran, on behalf of all authors

Reviewer 2 Report

Reviewer's comments

Title: Why a pilot study?

Introduction

The analysis of your references shows that they are not update. Very difficult to include a reference older than 5 years for this type of research.

2 :in the US [1].

Please, include a recent reference targeting the global impact.Not only in US.

54: Especially in children, the oral cavity is a highly heterogeneous ecological system.

Reference please, especially as this is debatable (elderly)

60: However, only a few studies have focused on the microbial profiles of tooth or site-specific supragingival plaque [33-36].

Be careful to distinguish between supragingival, occlusal, interdental biofilm, etc...

It seems , from reading your discussion, that studies have already been carried out?

The 2 references ( 1 or/and 2) could be added ? :

doi: 10.3390/microorganisms7090319

PLoS One. 2017 Oct 10;12(10):e0185804. doi: 10.1371/journal.pone.0185804. eCollection 2017.

77: Furthermore, all experiments were performed in accordance with relevant guidelines and regulations.

Which ones please?

Materials and Methods

You do not mention the STROBE guidelines. Please take this into account in the rewriting and include the completed checklist in the appendix

82 : Thirty healthy children were enrolled and stratified

This is not at all clear. The procedure must be clarified. Include a flow chart.

90: periodontitis

Are you sure? At this age?

91 : reduced saliva production,..

On what criteria ?

93 : radiographic exams

Why, as long as you use the DMFT index

99: The dental plaque collection was standardized, and all participants were asked to refrain from any oral hygiene activity, eating and drinking in the morning before oral sampling.

At what time did you collect the samples?

DMFT

It would have been more appropriate to use the ICDAS index, which is clinically much more refined and accurate.

Table 1 is well constructed. However, it would be helpful to break down DMFT upstream into D, M, and F. We have no idea what the distribution of these indices is.

Does an F tooth have the same probability of having a different quantity and quality of biofilm than a non-carious tooth with an equivalent occlusal anatomical profile

Was having at least one M molar (i.e. because of decay) an exclusion criterion?

How did you pool when at least one out of 4 teeth was missing?

137: A total of 36 occlusal plate samples: primary Dprim's (n=10), Dmixed's (n=8), and permanent 6mixed's (n=8), 6perm's (n=10) first molars were selected

Why was this done? What is the rationale and methods used?

147 : Shannon index

Reference please

Table 3: My advice is to remove it. Your text 176-179 is explicit enough.

Discussion

272: Moreover, caries risk estimation is performed for the entire oral cavity and does not represent the caries susceptibility of individual teeth.

Please review the bibliography. This is not correct

298 & 304: (Figure 2b). (Figure 3),

Do not put in the text of a discussion

312 : disease

Diseases ?

298 & 304 & 333 & 343: to other studies, previous investigations , previous studies, to other studies, which utilized pooled dental plaque

See my comments in the introduction

350 : would support our findings

Support ... or not ????

Please to Include the strengths and weaknesses of your study

References

See my previous comments

References 32 and 37 are not the same?

Conclusion of the reviewer

This submission is interesting, the section "bacteriological methods^ and the results are correct.

The problem++ lies in the methodology as described. I sincerely don't know if this is recoverable

The comments related to the introduction and references should be resolved without too much difficulty

One should also try to synthesize the discussion by trying not to repeat the results directly or indirectly.

Reviewer 3 Report

First of all, very interesting, and compliments for methodology and analysis - all is quite thorough.

My first concern is the lack of information about possible fluoride supplementation in children and use of mouthwashes. This information is missing and I believe it is necessary to include that information since the ingredients can affect the plaque composition.

The information related to Tables and Figures is redundant - you have so many details and they are already included in the text.

I have concern regarding this one subject (Table 3) who does not brush his/her teeth at all - somehow that person should not be included in the investigation since it means a great deal if someone brushes or does not brush at all. Statistics-wise, I would exclude that child.

Furthermore, since you already have so much data, it would be very interesting to correlate the microbial setup with specific oral hygiene habits - that would make sense.

Line 339 - you are missing the word "it" in the sentence beginning with "In the future..."

Overall, great job!
